# Clinico-Pathologic Profile of a Cohort of Patients with Actinic Keratosis in a Tertiary Center in Romania

**DOI:** 10.3390/cancers17121923

**Published:** 2025-06-10

**Authors:** Cristina Soare, Elena Codruța Cozma, Andrei Ludovic Poroșnicu, Daniel Alin Cristian, Draga Maria Mandi, Călin Giurcăneanu, Vlad Mihai Voiculescu

**Affiliations:** 1Department of Dermato-Oncology, Faculty of Medicine, “Carol Davila” University of Medicine and Pharmacy, 050474 Bucharest, Romania; cristina.vajaitu@drd.umfcd.ro (C.S.); calin.giurcaneanu@gmail.com (C.G.); vlad.voiculescu@umfcd.ro (V.M.V.); 2Department of Dermatology, “Elias” University Emergency Hospital, 011461 Bucharest, Romania; 3Department of Plastic Surgery, Faculty of Medicine, “Carol Davila” University of Medicine and Pharmacy, 050474 Bucharest, Romania; andrei.porosnicu@umfcd.ro; 4Department of Plastic Surgery, “Elias” University Emergency Hospital, 011461 Bucharest, Romania; 5Department of General Surgery, “Carol Davila” University of Medicine and Pharmacy, 050474 Bucharest, Romania; daniel.cristian@umfcd.ro; 6Department of General Surgery, Coltea Clinical Hospital, 030167 Bucharest, Romania

**Keywords:** actinic keratosis, histological subtypes, epidemiology, squamous cell carcinoma

## Abstract

Actinic keratoses are the most common precancerous lesions in the elderly population, and can sometimes be misdiagnosed as squamous cell carcinoma, which requires a different, often invasive, management, with possible risks of local complications. Our retrospective study analyzes the demographic, clinical and histopathological profile of a group of 58 patients with actinic keratoses surgically excised in the plastic surgery department of a tertiary center in Bucharest, Romania, highlighting the heterogeneity of histological subtypes, as well as the need for clinical pathological correlations in order to stratify patients into risk groups and to facilitate the choice of the best therapeutic option. Furthermore, the study highlights the continued need for early detection, the use of adequate photoprotection and the monitoring of patients with comorbidities that may increase the risk of developing AK/squamous cell carcinoma in situ.

## 1. Introduction

Actinic keratosis (AK) typically arises on skin that has been chronically exposed to ultraviolet radiation and is recognized as a component of the progressive pathological continuum leading to cancer development [1]. Within this framework, AK lesions are considered early forms of in situ squamous cell carcinoma (SCC), possessing the potential to progress into invasive malignancies and ultimately metastasize [2,3,4,5]. The estimated annual risk of progression from an individual AK lesion to invasive SCC ranges from 0.025% and 16% per year [6]. Notably, one study reported that approximately 27% of cutaneous SCCs originate from AK lesions, while 56% are associated with AK in proximity [7]. Another investigation found that approximately 65% of all primary SCCs and 36% of all primary basal cell carcinomas developed from lesions previously diagnosed as AKs [8]. These findings underscore the importance of early identification and management of AK.

AKs are the most common form of in situ carcinoma in humans [9]. Epidemiological studies estimate the prevalence of AK to vary between 6 and 26% in the general population [3,10]. Higher rates of AK prevalence are observed in regions with intense ultraviolet (UV) exposure. For instance, in Australia, the country with the highest incidence of skin cancer globally, the prevalence of AK among adults varies from 40 to 60% [11]. Studies estimate that approximately 8% of AKs progress to invasive SCC [12].

The occurrence of AK involves numerous factors, each of which acts through different molecular mechanisms. The molecular mechanisms mainly involve mutations of the TP53 gene, induced by UV radiation, which affects cell cycle control and apoptotic processes. Also, alterations of pathways involved in cell proliferation and survival (RAS/RAF/MAPK and PI3K/AKT) have also been described. At the same time, the oxidative stress—chronic inflammation continuum contributes to genomic instability, and the impairment of DNA repair mechanisms accentuates the accumulation of mutations [13].

SCC is the second most common non-melanoma skin cancer (NMSC), following basal cell carcinoma (BCC), with a globally rising incidence [14]. Compared to BCC, SCC has a higher metastatic potential and therefore a greater associated mortality rate, early recognition and management being essential to prevent neoplastic progression [15,16].

While histopathological examination remains essential for confirmation in clinically atypical or suspicious cases, the diagnosis of AK is generally made clinically or dermoscopically, with biopsy reserved for lesions unresponsive to treatment or those with features suggestive of invasive disease [17].

Clinical and dermoscopic differentiation is often challenging when it comes to AKs, in situ SCCs, invasive SCCs, and their various subtypes. In patients presenting with such lesions, early and prompt detection is essential for initiating effective treatment, which remains one of the most critical factors for long-term survival and quality of life [2,17].

Biopsy is recommended in cases where lesions present with atypical characteristics, such as large dimensions, ulceration, local symptoms (pruritus), bleeding, or otherwise clinically suspicious [1,18]. Beyond the lesions that are visibly detectable during clinical examination, a substantial number of AKs may exist in a subclinical form, particularly within sun-damaged skin. For example, confocal microscopy evaluation applied to adjacent skin that appears unaffected may uncover cellular and nuclear abnormalities indicative of early AK pathology [19].

The concept of field cancerization was initially introduced by Slaughter et al. in 1953 to describe histologically abnormal tissue adjacent to primary oral squamous cell carcinoma (SCC) [20]. This model was proposed to explain the frequent occurrence of multiple primary tumors and local recurrences within a contiguous area of tissue. Since its inception, advancements in molecular biology have substantiated the genetic underpinnings of this phenomenon across a wide spectrum of malignancies, including cutaneous cancers [21].

Field cancerization is characterized by the emergence of a clonal population of genetically altered, premalignant cells arising from a single progenitor cell. This initial cell undergoes a series of somatic mutations following exposure to carcinogenic stimuli. Over time, the progeny of this cell expands to occupy a larger area of tissue, creating a “field” of genetically similar, yet non-invasive, atypical cells. Additional genetic and epigenetic alterations within this field may ultimately give rise to subclones with invasive and metastatic potential, culminating in the development of carcinoma [21,22].

In the context of cutaneous malignancies, chronic exposure to UV radiation—particularly UVA and UVB wavelengths—constitutes the primary carcinogenic insult. UV-induced DNA damage, including the formation of cyclo-butane pyrimidine dimers and 6-4 photoproducts, can initiate the mutational cascade that leads to field cancerization [23]. The field cancerization paradigm therefore has significant implications for both surveillance and treatment strategies in patients with cutaneous malignancies, highlighting the need for comprehensive approaches that address not only the visible lesion but also the surrounding at-risk tissue [24,25].

Rowert-Huber et al. proposed a widely adopted histopathological grading system for AKs, which stratifies lesions according to the vertical distribution of atypical keratinocytes within the epidermis [2,26]. In grade AK I (KIN I—mild dysplasia), atypical cells are confined to the basal and immediate suprabasal layers, occupying roughly the lower third of the epidermis. Grade II (KIN II—moderate dysplasia) is characterized by atypical keratinocytes extending through the lower two-thirds of the epidermal layer. Grade III (KIN I—severe dysplasia) demonstrates full-thickness epidermal atypia (Figure 1) [2,26]. However, the risk of progression to SCC is not linear, directly proportional to the KIN grade and up to now it is not possible to assess the exact evolution of an AKs, although some histological aspects (presence of acantholysis and basal hyperproliferation) are associated with a higher risk of progression. Thus, the risk of malignant transformation, although lower, exists even for AKs KIN I, hence the need to follow and monitor carefully all patients diagnosed with AKs [27,28].

## 2. Materials and Methods

### 2.1. Study Design

A retrospective and descriptive analysis of the patients with confirmed histopathologic diagnosis of Ak, based on the histopathologic analysis of all excision specimens between 1 January 2018 and 31 December 2023 at the Department of Plastic Surgery, Elias Emergency University Hospital, Bucharest, Romania, was performed. Complete clinical and pathological data were collected and reviewed.

This study includes only cases of AK that were surgically excised and referred to the plastic surgery department. Non-excised lesions or those managed by dermatological means without histopathology confirmation were not available for analysis. Therefore, the total number of AKs diagnosed or treated during the study period is not known. Although standard destructive methods represent the gold standard for treatment of AKs, the decision for complete surgical excision is a “center” characteristic influenced by multiple factors: the first presentation of patients in the Plastic Surgery Department, the absence of dermoscopic evaluation, and the absence of patients adherence to follow-up visits, which led to complete excision of all clinically suspicious lesions by the plastic surgeon.

Excisional biopsies showing incomplete excision of the lesion and punch biopsies were excluded (Figure 2). The suspected diagnosis and incomplete clinical–pathological data were not collected.

Clinical data such as age, sex, anatomical site of the lesion, and comorbidities, and parameters related to the macroscopic aspect of the lesion, diameter of the lesion, and presence or absence of ulceration, were obtained from the clinical referral forms. Histopathological parameters, such as histologic subtype, AK grade, mitotic activity, cytonuclear pleomorphism, parameters related to the inflammatory infiltrate—subtype and severity, and parameters related to the perilesional skin—severity of the solar elastosis, were collected from the biopsy forms.

### 2.2. Data Analysis

Data analysis was performed using the IBM SPSS Statistics for Mac, Version 30.0 (Released 2024; IBM Corp., Armonk, NY, USA). The median for age was calculated, whereas frequencies and percentages for all other clinicopathological variables were evaluated. A *p*-value of <0.05 was considered significant. Fisher’s exact tests were applied to determine the association of various clinicopathological features concerning the histological subtypes.

### 2.3. Ethical Aspects

This study was conducted with the approval of the Ethics Committee of the Elias Emergency University Hospital, Bucharest, Romania (Approval no. 1179/25.02.2025). Patients’ rights were upheld in accordance with the World Health Organization’s guidelines and the Declaration of Helsinki.

## 3. Results

### 3.1. Demographic Data

A total of 58 cases of AKs were included in the present study. Of these, 26 patients were female, representing 44.8% of the cohort, while 32 were male, accounting for 55.2%. The age of patients ranged from 55 to 90 years, with a median age exceeding 77 years, indicating that AK predominantly affects individuals in the older age spectrum. A modest sex-related difference in incidence was observed, with male patients being more frequently affected than females—32 cases in men compared to 26 in women.

The overall mean age at diagnosis among the study population was 76.36 years (standard deviation [SD] = 7.84, 95% CI [74.3; 78.42]), suggesting that the condition predominantly affects older individuals. A notable age difference was observed between sexes, with a 2.2-year higher mean age among female patients compared to their male counterparts.

In terms of sex-specific age distribution, the mean age of female patients was 77.58 years (SD = 7.5), with a 95% confidence interval (CI) ranging from 74.55 to 80.61 years. The age span for this group extended from 61 to 90 years, indicating that female patients were generally diagnosed later in life. Conversely, male patients had a mean age of 75.38 years (SD = 8.08), with a 95% CI of 72.46 to 78.29 years, and an age range between 55 and 88 years.

An analysis of the age distribution of AK cases across age decades reveals a clear trend correlating increasing age with higher incidence. Out of the total 58 cases included in the study, only 2 cases (3.4%) were identified in individuals under the age of 60, indicating a very low frequency in younger populations. The most affected age group was between 70 and 79 years, comprising 24 cases (41.4%), followed closely by the 80–89 age group, with 22 cases (37.9%). The 60–69 age group accounted for 9 cases (15.5%). All patients included in the study had Fitzpatrick skin phototype III, which is the most common one in Romania.

Among the initial clinical diagnoses recorded, cutaneous squamous cell carcinoma (cSCC) was the most frequently suspected condition, documented in 19 of 58 cases (32.75%). Of these, eight cases were specifically categorized as carcinoma in situ, reflecting the difficulty in distinguishing early invasive or pre-invasive malignancies from AK based solely on gross morphology. In addition, basal cell carcinoma (BCC) was initially suspected in 12 cases (20.68%), further demonstrating the clinical similarity between various keratinocyte-derived lesions.

The remaining 10 lesions were initially diagnosed as a variety of benign skin conditions, including seborrheic keratosis (3 cases, 5.17%), verruca vulgaris or common warts (3 cases, 5.17%), cutaneous horn (2 cases, 3.44%), and Bowen’s disease (2 cases, 3.44%). These findings underline the considerable heterogeneity in the clinical presentation of AK and the potential for misclassification with both benign and malignant entities.

An analysis of the distribution of AK cases according to lesion size reveals a higher incidence of smaller lesions. Most patients (*n* = 31 cases), representing 53.45% of the total) presented with lesions measuring less than 1 cm in diameter. Lesions ranging between 1 cm and 2 cm were observed in 18 patients (31.03%), while those measuring between 2 cm and 3 cm were identified in 9 patients (15.52%).

Taken together, lesions equal to or larger than 1 cm were present in 27 patients (46.55%), an approximately equivalent proportion to that of smaller lesions within the study population. Smaller lesions, measuring less than 1 cm in diameter, were more frequently observed in male patients, accounting for 19 cases (61.3% of all lesions in this size category), compared to 12 cases (38.7%) in female patients. However, when considering lesions greater than or equal to 1 cm, the distribution between sexes was relatively balanced. Female patients presented with 14 such lesions (51.9%), while male patients had 13 lesions (48.1%).

The analysis of lesion frequency according to anatomical sites reveals that the head and neck region is the most affected, accounting for 51 out of 58 cases (87.9%). This is followed by the extremities, with six cases (10.3%), and the trunk, which was involved in only a single case (1.7%). Many lesions, regardless of histological subtype, were located on the head and neck region, with a statistically significant association observed (*p* = 0.008). Specifically, hypertrophic AKs were predominantly found on the head and neck (33 out of 35 cases, 94.3%), with only two lesions (5.7%) located on the extremities. Similarly, atrophic AKs were mainly localized to the head and neck in 13 of 16 cases (81.3%), while the remaining 3 lesions (18.8%) were found on the limbs. All three lichenoid AKs were exclusively identified in the head and neck region (100%).

In contrast, the Bowenoid subtype demonstrated the most balanced anatomical distribution: two out of four lesions (50%) were located on the head and neck, while one lesion each (25%) was found on the trunk and the extremities, respectively. This wider distribution pattern may reflect the distinct biological behavior or etiologic factors underlying Bowenoid lesions compared to other AK subtypes. The presented data are exposed in Table 1.

### 3.2. Histological Analysis

The analysis of histological subtypes indicates that hypertrophic lesions are the most frequently encountered form of AKs, identified in 35 out of 58 cases (60.3%). This was followed by atrophic lesions, observed in 16 cases (27.6%), Bowenoid lesions in four cases (6.9%), and lichenoid variants in three cases (5.2%). When examining the correlation between AK histological subtypes and lesion size, a statistically significant association was again observed (*p* = 0.05). Hypertrophic and lichenoid AKs were predominantly associated with lesions measuring less than 1 cm in diameter—24 out of 35 hypertrophic lesions (68.6%) and 2 out of 3 lichenoid lesions (66.7%) fell within this size category. In contrast, atrophic and Bowenoid AKs were more frequently associated with intermediate-sized lesions, measuring between 1 and 2 cm. Specifically, 8 out of 16 atrophic lesions (50%) and 3 out of 4 Bowenoid lesions (75%) were within this intermediate range. Among the histological subtypes identified, no acantholytic variants were observed in our cohort.

In assessing the degree of dysplasia, the Keratinocyte Intraepidermal Neoplasia (KIN) grade serves as the most widely accepted histopathological scoring system for quantifying the severity of atypia in AK lesions.

In the present cohort, analysis according to KIN grade demonstrated that moderate dysplasia (KIN II) was the most common, accounting for 24 cases (41.4%). This was followed by mild dysplasia (KIN I), identified in 20 patients (34.5%), and severe dysplasia (KIN III), present in 14 cases (24.1%) (Figure 3 and Figure 4). These findings suggest that a significant proportion of lesions exhibit moderate to advanced histopathological changes, underscoring the importance of early recognition and appropriate management of AK to prevent malignant progression. Table 1 demonstrates the relationship between various clinicopathological factors and the KIN grades of AKs. A significant association of KIN grades of AKs was noted with sex, cytonuclear pleomorphism, solar elastosis, and peritumoral inflammatory cell infiltrate severity. We found that KIN I and KIN III were more prevalent in the age group 70–79 years (40% and 42.9%, respectively), whereas KIN II was found to affect the most patients between 80 and 89 years of age (45.8%). KIN I and KIN III predominantly affected male patients (60% and 78.6%, respectively), whereas KIN II affected females (62.5%), thus associating a statistically significant difference (*p* = 0.042).

Regarding the association between AKs and solar elastosis, all 58 lesions included in the study demonstrated the presence of elastotic changes, albeit with varying degrees of severity. Marked solar elastosis was the most frequently observed, identified in 28 out of 58 lesions (48.3%). This was followed by moderate elastosis in 22 cases (37.9%), while mild elastotic changes were observed in only 8 lesions (13.8%). When examining the correlation between KIN grade and the severity of solar elastosis, a clear trend emerges: higher KIN grades are associated with more pronounced elastotic changes. Specifically, KIN I lesions were predominantly linked to mild solar elastosis, observed in 8 out of 20 cases (40%). In contrast, KIN II lesions were most frequently associated with moderate elastosis (14 out of 24 cases, 58.3%), while KIN III lesions were exclusively associated with severe solar elastosis.

Cytonuclear pleomorphism observed within the study cohort demonstrated a relatively balanced distribution between mild and moderate grades. Specifically, moderate pleomorphism was identified in 28 out of 58 lesions (48.3%), while mild pleomorphism was present in 24 lesions (41.4%). Severe pleomorphism represented the least frequent category, observed in only six lesions (10.3%). Further analyzing the cytonuclear pleomorphism according with KIN grade, we observed that most KIN I lesions were associated with mild pleomorphism (19 out of 20 cases, 95%). In contrast, lesions graded as KIN II and KIN III predominantly exhibited moderate pleomorphism, found in 19 of 24 cases (79.2%) and 8 of 14 cases (57.1%), respectively. This distribution reflects a statistically significant correlation between the degree of dysplasia and the severity of pleomorphism (*p* < 0.001), suggesting that increasing KIN grade is associated with greater cytonuclear atypia.

Regarding the peritumoral inflammatory infiltrate, its evaluation was performed both in terms of severity and the type of inflammatory cells present. Most cases exhibited a moderately severe peritumoral inflammatory infiltrate, observed in 33 out of 58 lesions (56.9%). This was followed by lesions with a marked inflammatory infiltrate in 17 cases (29.3%) and a mild infiltrate in 8 cases (13.8%). Across all KIN grades, the predominant type of peritumoral inflammatory infiltrate was lymphoplasmacytic. This pattern was consistently observed in 18 out of 20 KIN I lesions (90%), 22 out of 24 KIN II lesions (91.7%), and 12 out of 14 KIN III lesions (85.7%), indicating that the nature of the infiltrate remains relatively stable regardless of dysplasia severity. However, when analyzing the severity of the peritumoral inflammatory response in relation to KIN grade, a clear pattern emerges. KIN I and KIN II lesions were most frequently associated with a moderate infiltrate, identified in 10 of 20 cases (50%) and 18 of 24 cases (75%), respectively. In contrast, KIN III lesions were predominantly associated with a severe inflammatory infiltrate, observed in 9 out of 14 cases (64.3%). Importantly, mild inflammatory infiltrates were found exclusively in KIN I lesions (8 of 20 cases, 40%) and were absent in higher-grade lesions. These findings suggest a statistically significant association between increasing KIN grade and the severity of the peritumoral inflammatory infiltrate (*p* < 0.001). Many AK lesions in the study cohort did not exhibit ulceration, with 40 out of 58 cases (69%) showing an intact surface epithelium. The presented data are exposed in Table 1, Table 2 and Table 3.

### 3.3. Associated Comorbidities Analysis

In evaluating patient comorbidities, we assessed the prevalence of smoking, the presence of cardiovascular diseases, diabetes mellitus, and a history of neoplastic conditions (Table 1). Among the 58 patients included in the study, only 8 individuals (13.8%) were recorded as smokers. However, this percentage may be underestimated, as smoking status may not have been consistently documented for all patients. Notably, four of the smokers presented with AKs located on the lips, while the remaining four had lesions in other areas of the head and neck region. Of the total six cases of lip-localized AK, four (66.7%) were associated with smoking, which is consistent with the existing literature that identifies tobacco use as a contributing factor in the development of actinic damage, particularly in the perioral area.

Cardiovascular comorbidities, including conditions such as hypertension and heart failure, were observed in 35 out of 58 patients (60.3%), a finding that is likely related to the advanced age of individuals affected by AKs. Conversely, diabetes mellitus was identified in only 8 patients (13.8%), representing a lower proportion of the cohort.

Additionally, a relatively high number of patients (11 out of 58; 19%) had a documented history of neoplastic disease. This may be explained by the increased susceptibility of oncology patients to AKs due to immunosuppressive treatments, cumulative sun exposure, or shared etiologic factors. Furthermore, the elevated rate of neoplasia in this group may be influenced by the hospital’s profile, which includes a dedicated oncology department, thereby potentially increasing the representation of such cases within the study population. Detailed data are presented in Table 1, Table 2 and Table 3.

The results may reflect a progressive enhancement of the host immune response with increasing degrees of epithelial atypia, potentially serving as an indirect marker of lesion severity. This is consistent with prior studies suggesting that more dysplastic or advanced intraepidermal neoplasms can elicit stronger local immune reactions due to increased antigenicity or epithelial disruption. None of the patients included in the study were taking known photosensitizing medications such as thiazide diuretics or azathioprine.

## 4. Discussion

This comprehensive retrospective analysis offers a detailed clinicopathological profile of AKs in a Romanian cohort over a six-year period. The study reinforces the multifactorial etiology of AK, highlighting the interplay between chronic UV exposure, patient-related factors (age, sex, comorbidities), and lesion-specific histopathologic features.

The variability in clinical diagnosis of AKs and related lesions underscores the critical need for thorough evaluation using non-invasive diagnostic tools prior to considering surgical interventions. Dermoscopy has proven to significantly enhance diagnostic accuracy for AKs, aiding clinicians in distinguishing between benign and malignant lesions and thereby reducing unnecessary excisions. In instances where dermoscopic findings are inconclusive, RCM serves as a valuable adjunct, offering detailed cellular-level imaging that can further clarify diagnoses and potentially obviate the need for biopsy. Consequently, it is imperative that patients undergo comprehensive dermatologic and dermoscopic assessments before any surgical procedures are performed by plastic surgeons. This approach ensures that only lesions with a high index of suspicion for malignancy are excised, optimizing patient care by minimizing unnecessary excisions.

### 4.1. Epidemiology-Related Aspects

Consistent with previous studies, AK was predominantly observed in elderly patients, with a median age of 76.36 years and a peak incidence in the seventh and eighth decades of life. This finding supports the role of cumulative lifetime UV exposure as a key pathogenic driver of AK [29,30].

A higher prevalence of AK among men was reported, thus this pattern aligns with existing epidemiological data. This disparity has been attributed to several contributing factors, including greater cumulative ultraviolet (UV) exposure due to occupational and recreational behaviors, as well as generally lower adherence to photoprotective practices among men. The observed male predominance in this cohort therefore reinforces previously documented trends and highlights the ongoing need for targeted public health interventions, particularly among high-risk populations [30,31]. Androgenic alopecia is more prevalent in men than in women, which, combined with men’s generally shorter hair, results in less natural protection of the scalp and regions of the face who have a sun protection conferred by the hair (such as ears or temporal area) from UV radiation. Hair serves as an important photoprotective barrier, and clinical evidence confirms that individuals with balding or thinning hair have a higher risk of developing UV-induced precancerous lesions on the exposed scalp [32]. This mechanism likely underlies the higher incidence of AKs observed in male patients compared to female patients.

The findings of our study suggest a slightly earlier age of onset or clinical detection in men, which may be attributed to differences in occupational sun exposure, lifestyle habits, or more consistent and effective use of photoprotective measures reported among women, which could potentially delay the onset or clinical manifestation of sun-related skin conditions [33]. Together, these data underscore the importance of considering sex-specific factors in both the prevention and early detection of actinic skin lesions, as well as the potential influence of behavioral patterns on disease presentation.

Regarding the medium size of lesions at diagnosis, our study suggests that while AKs are more frequently diagnosed at smaller sizes, a substantial percentage still present with larger dimensions. This may reflect variations in the timing of diagnosis, lesion growth dynamics, or differences in patient awareness and access to dermatologic care. The comparable distribution between small and larger lesions also underscores the need for early detection and continuous monitoring, as lesion size may be associated with increased risk of progression to invasive squamous cell carcinoma [15].

A higher incidence of smaller AK lesions among men was found in the current study, which may be attributed to earlier detection, differing patterns of sun exposure, or variations in skin type and response to UV radiation. In contrast, the nearly equal distribution of larger lesions between sexes indicates that, despite behavioral or physiological differences, the progression to more extensive lesions occur at similar rates in both genders. This reinforces the importance of regular dermatological screening in both men and women, regardless of lesion size, to enable early identification and management.

Regarding the area of residence, with a predilection for urban area, may be partially explained by the ongoing demographic shift observed in recent years, characterized by increased migration from rural to urban areas. Additionally, the expansion of metropolitan areas has led to the administrative incorporation of peripheral rural communities into urban territories. As a result, some patients who were previously considered rural residents are now officially classified as urban dwellers. This urban predominance in the sample may therefore reflect both population movement and reclassification of geographic boundaries, rather than a true epidemiological disparity in disease occurrence between urban and rural settings.

The anatomical distribution of lesions strongly favored sun-exposed regions, particularly the head and neck (87.9%), which is in line with multiple studies demonstrating the photodistribution pattern of AK [34,35]. These findings are consistent with the established association between AKs and chronic UV exposure, as the head and neck are typically the most sun-exposed areas of the body. The relatively low involvement of the trunk and limbs may reflect both reduced sun exposure and possible differences in patient detection or reporting.

### 4.2. Histopathology-Related Aspects

The universal presence of solar elastosis in our cases further underscores the chronic photodamage underlying these lesions, with more severe elastosis correlating significantly with higher KIN grades (*p* < 0.001), echoing findings from Schmitz and others who identified solar elastosis as a surrogate marker of cumulative UV-induced dermal degeneration and lesion progression [35]. Solar elastosis, a histopathological hallmark of long-term photoaging, reflects the cumulative damage inflicted by UV exposure on dermal connective tissue. Its universal presence among the lesions examined reinforces the concept that UV-induced structural and molecular alterations in the skin microenvironment are central to the development and progression of AKs.

From a histopathological standpoint, hypertrophic AKs were the most common subtype, in accordance with the previous literature that identifies this variant as a prevalent and potentially more conspicuous form of AK [2]. Each subtype showed characteristic patterns of anatomical distribution and lesion size—an observation supported by studies suggesting that different AK subtypes may represent distinct biological stages or morphologic expressions of a shared etiological process [36].

The analysis of dysplasia severity using the KIN grading system revealed a predominance of KIN II lesions, followed by KIN I and KIN III, in order of frequency. Increasing KIN grade was significantly associated with more severe cytonuclear pleomorphism, pronounced solar elastosis, and more intense peritumoral inflammation, findings that mirror results from Röwert-Huber et al. (2007) and from more recent analyses correlating KIN grade with molecular progression toward invasive SCC [2,34]. These parameters may serve as early histological indicators of malignant potential.

While the peritumoral inflammatory infiltrate was predominantly lymphoplasmacytic across all lesions—consistent with the known immune response to UV-damaged keratinocytes—its severity increased with dysplasia grade. This suggests a possible amplification of the host immune response in the presence of advanced epithelial atypia, which may have both diagnostic and prognostic relevance.

The relatively low prevalence of ulceration (31%) and the absence of this feature in most AK lesions are in keeping with its characterization as a superficial, intraepidermal lesion. As noted in previous work by Quaedvlieg et al. the presence of ulceration may raise suspicion for early SCC transformation and should prompt thorough histopathologic evaluation [29].

The evolution of KIN toward cSCC has traditionally been described through a classic sequential pathway, in which lesions progress in a stepwise manner from KIN I to KIN III, ultimately culminating in invasive SCC. However, emerging evidence supports an alternative differentiated pathway, suggesting that any KIN grade—regardless of its position within the traditional hierarchy—may serve as a precursor to cSCC. This alternative model emphasizes the importance of histopathological and clinical factors such as lesion size, presence of ulceration, degree of cytonuclear pleomorphism, basal hyperproliferation, presence/absence of ancatholysis and the status of the perilesional microenvironment [27,28].

In our study, we observed that ulceration (*p* = 0.524) and lesion dimensions (*p* = 0.172) occurred across all KIN grades without statistically significant correlation to the severity of intraepidermal neoplasia. These findings align with the differentiated pathway, supporting the notion that the potential for malignant transformation is not confined to higher-grade lesions alone.

Conversely, our analysis demonstrated statistical correlations between cytonuclear pleomorphism (*p* < 0.001), solar elastosis (*p* < 0.001), and the severity of peritumoral inflammatory cell infiltrate (*p* < 0.001) with an increasing KIN grade. These associations reinforce the validity of the classic pathway, in which progressive histological changes correspond with higher grades and a stepwise carcinogenic trajectory.

Taken together, our data suggest a dualistic model of progression, wherein both the classical and differentiated pathways may coexist. This underscores the heterogeneity of KIN lesions and the necessity of a multifactorial approach in assessing their malignant potential.

### 4.3. Associated Comorbidities

Regarding comorbidities, our cohort showed high rates of cardiovascular disease (60.3%), reflecting the older demographic affected by AK, and a noteworthy proportion of patients with prior neoplastic conditions (19%). These findings align with studies linking immunosuppression or oncologic history with an increased risk of AK and non-melanoma skin cancers [37]. Smoking, while likely underreported in our cohort, was associated with AKs located on the lips—a site commonly affected by tobacco-induced photodamage—further supporting the established link between smoking and mucocutaneous AKs [38].

Finally, the statistically significant associations between AK subtypes and both lesion size and anatomical location (*p* < 0.05 and *p* = 0.008, respectively) highlight the diagnostic importance of integrating morphological patterns with clinical context. This can inform both prognosis and treatment planning, especially in distinguishing high-risk lesions that warrant excision from those manageable through field-directed therapies.

## 5. Limitations

This study is not without its limitations. Firstly, the analysis was conducted within a single tertiary care center, which may limit the generalizability of the findings to broader populations or diverse geographic and clinical settings. In addition, the relatively small sample size of 58 patients may reduce the statistical power to detect more subtle associations or less common clinicopathological patterns of AKs.

Furthermore, due to the retrospective design of the study, longitudinal follow-up data regarding clinical outcomes, lesion progression, or recurrence could not be assessed. As such, the natural history of the lesions and the potential transition to invasive squamous cell carcinoma could not be evaluated. Another notable limitation is the absence of a structured analysis of patient-specific risk factors, such as immunosuppression status, occupational UV exposure, or genetic predisposition, which could provide additional insight into disease etiology and progression.

Therapeutic interventions and treatment outcomes were also not addressed in this study, limiting the ability to evaluate the efficacy of various management strategies or to correlate treatment approaches with histological subtypes or severity grades (e.g., KIN I–III). This gap underscores the need for integrated clinical–pathological research that includes both diagnostic and therapeutic dimensions.

Considering these limitations, future studies with a prospective design, larger and more heterogeneous patient populations, and multicenter participation are strongly recommended. Such investigations should aim to explore in greater depth the associations between clinicopathological features and AK severity, as well as assess outcomes related to treatment efficacy, recurrence, and progression. Incorporating molecular profiling and follow-up data could further enhance our understanding of the biological behavior of AK and its potential for malignant transformation.

## 6. Conclusions

In conclusion, this study adds to the growing body of literature, affirming that AKs represent not only a marker of chronic UV damage but also a biologically heterogeneous group of intraepidermal neoplasms with varying clinical and histological features. These differences may reflect divergent risks of progression to invasive SCC. Comprehensive clinical–pathological correlation, as demonstrated in this study, remains essential for accurate diagnosis, appropriate risk stratification, and the development of individualized treatment strategies.

Future research incorporating molecular and immunohistochemical profiling, as well as long-term follow-up data, will be critical in refining the classification of AK, improving prognostic precision, and guiding therapeutic interventions tailored to lesion subtype and severity.

## Figures and Tables

**Figure 1 cancers-17-01923-f001:**
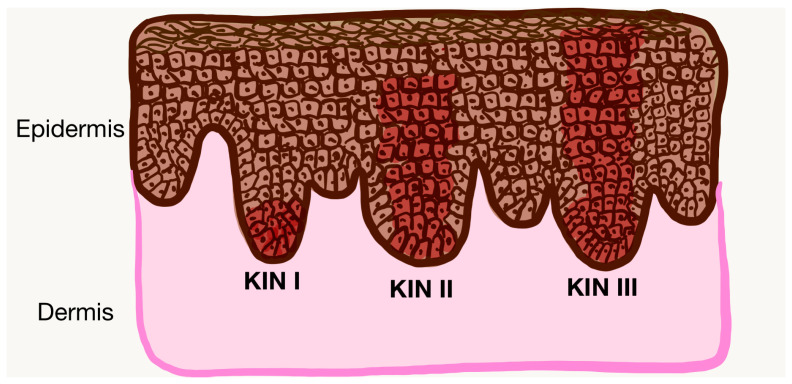
Histopathological grading of AKs (KIN I–III). Color coding: brown—normal keratinocytes, red—dysplastic keratinocytes in AK lesion. Image created in CanvaPro (https://www.canva.com/pro/, accessed on 1 May 2025).

**Figure 2 cancers-17-01923-f002:**
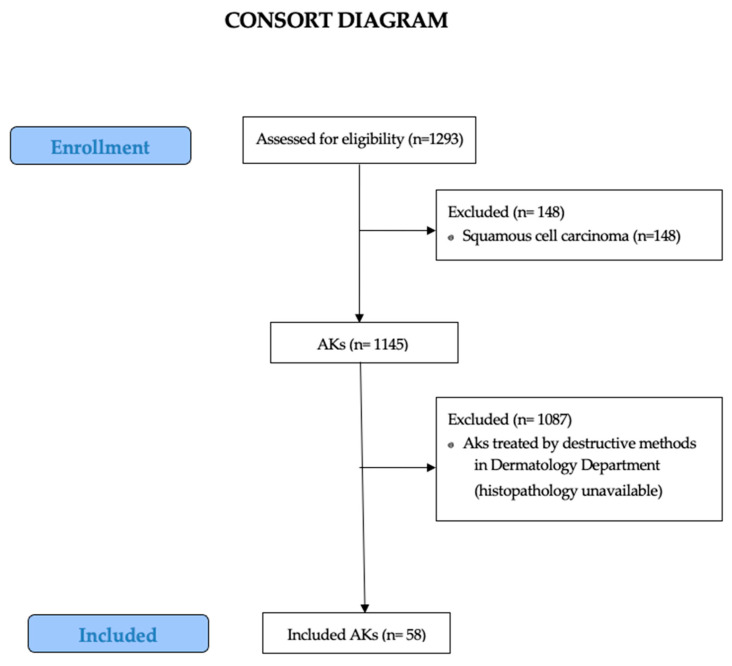
CONSORT diagram representing the selection process of the cases included in the study.

**Figure 3 cancers-17-01923-f003:**
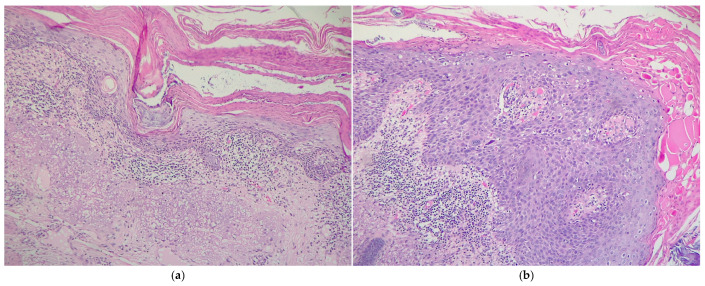
Histopathological aspects of (**a**) KIN I–II and (**b**) KIN III AKs. Magnification ×100; hematoxylin–eosin stain.

**Figure 4 cancers-17-01923-f004:**
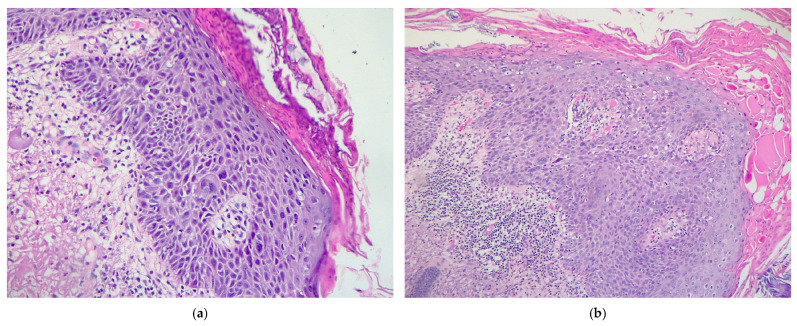
Histopathological aspects of (**a**) KIN II–III and (**b**) KIN III AKs. Magnification ×200; hematoxylin–eosin stain.

**Table 1 cancers-17-01923-t001:** Clinicopathological profile of study population.

Clinicopathologic Parameters	Values
**Age (years), median (IQR)**	77.0 (10.0)
**Age groups**	
50–59 years, *n* (%)	2 (3.4)
60–69 years, *n* (%)	9 (15.5)
70–79 years, *n* (%)	24 (41.4)
80–89 years, *n* (%)	22 (37.9)
>90 years, *n* (%)	1 (1.7)
**Gender**	
Male, *n* (%)	32 (55.2)
Female, *n* (%)	26 (44.8)
**Area of residence (urban vs. rural)**	
Urban, *n* (%)	38 (65.5)
Rural, *n* (%)	20 (34.5)
**Anatomical site**	
Head and neck, *n* (%)	51 (87.9)
Thorax, *n* (%)	1 (1.7)
Upper and lower limbs, *n* (%)	6 (10.3)
**Lesion dimensions**	
<1 cm, *n* (%)	31 (53.4)
1–2 cm, *n* (%)	18 (31)
2–3 cm, *n* (%)	9 (15.5)
**KIN grade**	
KIN I, *n* (%)	20 (34.5)
KIN II, *n* (%)	24 (41.4)
KIN III, *n* (%)	14 (24.1)
**Histopathological subtype**	
Hypertrophic, *n* (%)	35 (60.3)
Atrophic, *n* (%)	16 (27.6)
Lichenoid, *n* (%)	3 (5.2)
Bowenoid, *n* (%)	4 (6.9)
**Cytonuclear pleomorphism**	
Mild, *n* (%)	24 (41.4)
Moderate, *n* (%)	28 (48.3)
Severe, *n* (%)	6 (10.3)
**Solar elastosis**	
Mild, *n* (%)	8 (13.8)
Moderate, *n* (%)	22 (37.9)
Severe, *n* (%)	28 (48.3)
**Peritumoral inflammatory cell infiltrate severity**	
Mild, *n* (%)	8 (13.8)
Moderate, *n* (%)	33 (56.9)
Severe, *n* (%)	17 (29.3)
**Peritumoral inflammatory cell infiltrate type**	
Lymphoplasmacytic	52 (89.7)
Polymorphic	6 (10.3)
**Ulceration**	
Present, *n* (%)	18 (31)
Absent, *n* (%)	40 (69)
**Comorbidities**	
Smoking, *n* (%)	8 (13.8)
Cardiovascular, *n* (%)	35 (60.3)
Diabetes, *n* (%)	8 (13.8)
Cancer, *n* (%)	11 (19)

**Table 2 cancers-17-01923-t002:** Association of clinicopathological parameters with KIN grades of AKs.

Clinicopathological Parameters	KIN Grade			*p*-Value
	KIN I	KIN II	KIN III	
**Age group**				
50–59 years, *n* (%)	0 (0)	0 (0)	2 (14.3)	0.145
60–69 years, *n* (%)	5 (25)	3 (12.5)	1 (7.1)	
70–79 years, *n* (%)	8 (40)	10 (41.7)	6 (42.9)	
80–89 years, *n* (%)	7 (35)	11 (45.8)	4 (37.9)	
>90 years, *n* (%)	0 (0)	0 (0)	1 (7.1)	
**Gender**				
Male, *n* (%)	12 (60)	9 (37.5)	11 (78.6)	0.042 **
Female, *n* (%)	8 (40)	15 (62.5)	3 (21.4)	
**Area of residence (urban vs. rural)**				
Urban, *n* (%)	13 (65)	14 (58.3)	38 (65.5)	0.448
Rural, *n* (%)	7 (35)	10 (41.7)	20 (34.5)	
**Anatomical site**				
Head and neck, *n* (%)	18 (90)	20 (83.3)	13 (92.9)	0.229
Thorax, *n* (%)	0 (0)	0 (0)	1 (7.1)	
Upper and lower limbs, *n* (%)	2 (10)	4 (16.7)	0 (0)	
**Lesions dimensions**				
<1 cm, *n* (%)	15 (75)	9 (37.5)	7 (50)	0.172
1–2 cm, *n* (%)	3 (15)	10 (41.7)	5 (35.7)	
2–3 cm, *n* (%)	2 (10)	5 (20.8)	2 (14.3)	
**Histopathological subtype**				
Hypertrophic, *n* (%)	12 (60)	12 (50)	11 (78.6)	0.257
Atrophic, *n* (%)	7 (35)	8 (33.3)	1 (7.1)	
Lichenoid, *n* (%)	1 (5)	2 (8.3)	0 (0)	
Bowenoid, *n* (%)	0 (0)	2 (8.3)	2 (14.3)	
**Cytonuclear pleomorphism**				
Mild, *n* (%)	19 (95)	5 (20.8)	0 (0)	<0.001 **
Moderate, *n* (%)	1 (5)	19 (79.2)	8 (57.1)	
Severe, *n* (%)	0 (0)	0 (0)	6 (42.9)	
**Solar elastosis**				
Mild, *n* (%)	8 (40)	0 (0)	0 (0)	<0.001 **
Moderate, *n* (%)	7 (35)	14 (58.3)	1 (7.1)	
Severe, *n* (%)	5 (25)	10 (41.7)	13 (92.9)	
**Peritumoral inflammatory cell infiltrate severity**				
Mild, *n* (%)	8 (40)	0 (0)	0 (0)	<0.001 **
Moderate, *n* (%)	10 (50)	18 (75)	5 (35.7)	
Severe, *n* (%)	2 (10)	6 (25)	9 (64.3)	
**Peritumoral inflammatory cell infiltrate type**				
Lymphoplasmacytic	18 (90)	22 (91.7)	12 (85.7)	0.843
Polymorphic	2 (10)	2 (8.3)	2 (14.3)	
**Ulceration**				
Present, *n* (%)	5 (25)	7 (29.2)	6 (42.9)	0.524
Absent, *n* (%)	15 (75)	17 (70.8)	8 (57.1)	
**Comorbidities**				
Smoking, *n* (%)	3 (15)	2 (8.3)	3 (21.4)	0.519
Cardiovascular, *n* (%)	11 (55)	15 (62.5)	9 (64.3)	0.829
Diabetes, *n* (%)	3 (15)	2 (8.3)	3 (21.4)	0.519
Cancer, *n* (%)	6 (30)	4 (16.7)	1 (7.1)	0.230

** statistically signficant.

**Table 3 cancers-17-01923-t003:** Association of clinicopathological parameters with histopathological subtypes of AKs.

Clinicopathological Parameters	Histopathological Subtype				*p*-Value
	Hypertrophic	Atrophic	Lichenoid	Bowenoid	
**Age group**					
50–59 years, *n* (%)	2 (5.7)	0 (0)	0 (0)	0 (0)	0.981
60–69 years, *n* (%)	6 (17.1)	3 (18.3)	0 (0)	0 (0)	
70–79 years, *n* (%)	13 (37.1)	7 (43.8)	2 (66.7)	2 (50)	
80–89 years, *n* (%)	13 (37.1)	6 (37.5)	1 (33.3)	2 (50)	
>90 years, *n* (%)	1 (2.9)	0 (0)	0 (0)	0 (0)	
**Gender**					
Male, *n* (%)	21 (60)	10 (62.5)	0 (0)	1 (25)	0.120
Female, *n* (%)	14 (40)	6 (37.5)	3 (100)	3 (75)	
**Area of residence (urban vs. rural)**					
Urban, *n* (%)	23 (65.7)	9 (56.3)	2 (66.7)	4 (100)	0.438
Rural, *n* (%)	12 (34.3)	7 (43.8)	1 (33.3)	0 (0)	
**Anatomical site**					
Head and neck, *n* (%)	33 (94.3)	13 (81.3)	3 (100)	2 (50)	0.008 **
Thorax, *n* (%)	0 (0)	0 (0)	0 (0)	1 (25)	
Upper and lower limbs, *n* (%)	2 (5.7)	3 (18.8)	0 (0)	1 (25)	
**Lesion dimensions**					
<1 cm, *n* (%)	24 (68.6)	5 (31.3)	2 (66.7)	0 (0)	0.05 **
1–2 cm, *n* (%)	6 (17.1)	8 (50)	1 (33)	3 (75)	
2–3 cm, *n* (%)	5 (14.3)	3 (18.8)	0 (0)	1 (25)	
**KIN grade**					
KIN I, *n* (%)	12 (34.3)	7 (43.8)	1 (33.3)	0 (0)	0.257
KIN II, *n* (%)	12 (34.3)	8 (50)	2 (66.7)	2 (50)	
KIN III, *n* (%)	11 (31.4)	1 (6.3)	0 (0)	2 (50)	
**Cytonuclear polymoprhism**					
Mild, *n* (%)	5 (14.3)	0 (0)	0 (0)	1 (25)	0.299
Moderate, *n* (%)	17 (48.6)	7 (43.8)	1 (33.3)	3 (75)	
Severe, *n* (%)	13 (37.1)	9 (56.3)	2 (66.7)	0 (0)	
**Solar elastosis**					
Mild, *n* (%)	15 (42.9)	7 (43.8)	2 (66.7)	4 (100)	0.308
Moderate, *n* (%)	15 (42.9)	7 (43.8)	0 (0)	0 (0)	
Severe, *n* (%)	5 (14.3)	2 (12.5)	1 (33.3)	0 (0)	
**Peritumoral inflammatory cell infiltrate severity**					
Mild, *n* (%)	6 (17.1)	2 (12.5)	0 (0)	0 (0)	0.526
Moderate, *n* (%)	17 (48.6)	12 (75)	2 (66.7)	2 (50)	
Severe, *n* (%)	12 (34.3)	2 (12.5)	1 (33.3)	2 (50)	
**Peritumoral inflammatory cell infiltrate type**					
Lymphoplasmacytic	32 (91.4)	14 (87.5)	2 (66.7)	4 (100)	0.499
Polymorphic	3 (8.6)	2 (12.5)	1 (33.3)	0 (0)	
**Ulceration**					
Present, *n* (%)	10 (28.6)	6 (37.5)	1 (33.3)	1 (25)	0.922
Absent, *n* (%)	25 (71.4)	10 (62.5)	2 (66.7)	3 (75)	
**Comorbidities**					
Smoking, *n* (%)	5 (14.3)	2 (12.5)	0 (0)	1 (25)	0.818
Cardiovascular, *n* (%)	17 (48.6)	13 (81.3)	2 (66.7)	3 (75)	0.147
Diabetes, *n* (%)	6 (17.1)	2 (12.5)	0 (0)	0 (0)	0.689
Cancer, *n* (%)	7 (20)	2 (12.5)	1 (33.3)	1 (25)	0.812

** statisically significant.

## Data Availability

The original contributions presented in this study are included in the article. Further inquiries can be directed to the corresponding author.

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
