# Peer review of "Clinico-Pathologic Profile of a Cohort of Patients with Actinic Keratosis in a Tertiary Center in Romania"

_cancers, 2025, doi:10.3390/cancers17121923_

Round 1
Reviewer 1 Report (Previous Reviewer 1)
Comments and Suggestions for Authors
None
Reviewer 2 Report (Previous Reviewer 2)
Comments and Suggestions for Authors
The authors have been responsive.
This manuscript is a resubmission of an earlier submission. The following is a list of the peer review reports and author responses from that submission.
Round 1
Reviewer 1 Report
Comments and Suggestions for Authors
Specific Feedback for the Authors:
1. Introduction Section: A brief overview of the different variants of actinic keratosis (AK) and their underlying molecular mechanisms would strengthen the context of the study.
2. Figure Addition: A standalone figure illustrating KINI (Keratinocytic Intraepidermal Neoplasia) would enhance the manuscript’s educational value.
Author Response
Thank you for your comments. I have made the changes accordingly.
Reviewer 2 Report
Comments and Suggestions for Authors
This is an interesting study in which excised actinic keratoses (AK) were evaluated clinically and pathologically.
- It is not standard of care to excise AK. Freezing, curetting and other destructive modalities may be more commonly employed. This study may have been more likely to include AK with more atypical clinical features leading to excision as treatment. How do the authors explain this potential source of bias?
- Since I biopsy very few AK, it was revealing to see the variety of pathologic presentations.
- The results largely repeat the information in the tables. The results section can be shortened.
Author Response
Thank you for your valuable comments. Yes, in our dermatology department, the destructive methods are preferred. Nevertheless, in Romania the patients tend to present in great numbers in the plastic surgery department, where the treatments tend to be surgical without a dermatology evaluation before surgery. For that, the patients presented in this study had been managed by the plastic surgeon and allowed us a retrospective study that analyzes the clinico-pathological aspects of AK.
Reviewer 3 Report
Comments and Suggestions for Authors
Interesting article, given the originality of the subject, which is little discussed. There are some typos. It would also be interesting to include additional histological images also concerning the perilesional component, as well as clinical images in progression of actinic keratosis/carcinoma and clinical images of the histology presented.
Author Response
Thank you for your comment. We have added the clinical images. Regarding the perilesional component, the protocol in our hospital is to keep in the library only the slides with the lesional component, so we do not have access to other slides. Regarding the length of the Results section, we had removed some paragraphs that were already explained in the tables.